# Enhancing Medical Image Retrieval with UMLS-Integrated CNN-Based Text Indexing

**DOI:** 10.3390/diagnostics14111204

**Published:** 2024-06-06

**Authors:** Karim Gasmi, Hajer Ayadi, Mouna Torjmen

**Affiliations:** 1Department of Computer Science, College of Computer and Information Sciences, Jouf University, Sakaka 72388, Saudi Arabia; 2Information Retrieval and Knowledge Management Research Laboratory, York University, Toronto, ON M3J 1P3, Canada; hajaya1@yorku.ca; 3Research Laboratory on Development and Control of Distributed Applications (REDCAD), National Engineering School of Sfax, Sfax University, Sfax 3029, Tunisia; mouna.torjmen@redcad.org

**Keywords:** text-based medical image retrieval, Convolutional Neural Network, Medical-Dependent Features, UMLS metathesaurus

## Abstract

In recent years, Convolutional Neural Network (CNN) models have demonstrated notable advancements in various domains such as image classification and Natural Language Processing (NLP). Despite their success in image classification tasks, their potential impact on medical image retrieval, particularly in text-based medical image retrieval (TBMIR) tasks, has not yet been fully realized. This could be attributed to the complexity of the ranking process, as there is ambiguity in treating TBMIR as an image retrieval task rather than a traditional information retrieval or NLP task. To address this gap, our paper proposes a novel approach to re-ranking medical images using a Deep Matching Model (DMM) and Medical-Dependent Features (MDF). These features incorporate categorical attributes such as medical terminologies and imaging modalities. Specifically, our DMM aims to generate effective representations for query and image metadata using a personalized CNN, facilitating matching between these representations. By using MDF, a semantic similarity matrix based on Unified Medical Language System (UMLS) meta-thesaurus, and a set of personalized filters taking into account some ranking features, our deep matching model can effectively consider the TBMIR task as an image retrieval task, as previously mentioned. To evaluate our approach, we performed experiments on the medical ImageCLEF datasets from 2009 to 2012. The experimental results show that the proposed model significantly enhances image retrieval performance compared to the baseline and state-of-the-art approaches.

## 1. Introduction

Medical information retrieval has a range of applications and solutions connected with better health care. At a basic level, it encompasses image retrieval, retrieval of reports, and natural language queries to databases containing both images and text. However, image retrieval is a challenging task as it can be very subjective, requiring high-level cognitive processing. There are two main types of image retrieval used clinically. One is where the medical professional has a clear idea of what they are looking for and uses the image to seek specific information. The second is a case in which the medical professional has an image and desires to find all similar images to aid diagnosis or as a teaching aid. Step one carries manual work, as tags need to be manually attached to the image usually as metadata. When the images are stored in large databases such as Picture Archiving and Communication Systems (PACS), this can be a highly disorganized and time-consuming task. Step two involves searching using the image as the query and algorithmic methods using visual features of the image attempt to retrieve similar images. As technology advances, there has been increasing support to move to automatic image annotation and content-based retrieval. It is within content-based image retrieval (CBIR) that the model CNNMIR seeks to improve the current state of the art.

Usually in medical domain, images constitute a reference set of previously evaluated cases, that physicians may use to make the right decisions. With the massive growth of medical images, it becomes hard for domain experts to find relevant images in large medical datasets. Thus, the need for an efficient and effective medical image retrieval system becomes urgent [1]. Two main approaches for medical image retrieval are widely used: text- and content-based retrievals. These approaches search for relevant images by using different principles: the text-based approach relies on the high-level semantic features of the images, however, the content-based approach relies on the low-level visual features (e.g., color, shape, and texture) of the image. Comparing both approaches, the Content-Based Medical Image Retrieval (CBMIR) performance is less favorable due to the gap between low-level visual features and high-level semantic features [2,3]. Therefore, several medical image retrieval systems apply the Text-Based Medical Image Retrieval (TBMIR) approach to search for images [4]. Most of these approaches are: either, traditional simple keyword-based approaches; where the meanings of medical entities are ignored, or concept-based approaches; that are time and disk-space consuming. According to our previous works [5,6], the presence of specific medical information, namely Medical-Dependant Features (MDF) in the textual descriptions of medical images has a positive impact on the performance of TBMIR approaches.

In these last years, Convolutional Neural Network (CNN) [7,8] models have shown significant performance improvement in several fields as Natural Language Processing (NLP) [9] and computer vision [10]. Given their success in such fields, it seems to be efficient for image retrieval. Unfortunately, until now, the CNN models have not a significant positive impact on medical image retrieval, especially on text-based medical image retrieval (TBMIR) [11]. It may be due to the complexity of the ranking process: it is not obvious how to consider TBMIR tasks as an image retrieval task [12] and not as a traditional information retrieval task, nor an NLP task. Indeed, the traditional information retrieval systems identify the relevance of a document to a given query; however, the NLP systems deduce the semantic relations between the query and the document. These two systems do not take into account the specificity of images in their processes.

In our previous work [13], we proposed a personalized CNN model that considers the specificity of images in its retrieval process. In that model, we consider the Word2Vec model for word embedding. However, it is well known that the Word2Vec model considers general terminologies, which are not specific for any domain. As our work fits in the medical image retrieval field, we believe that using medical semantic resources, such as UMLS, for converting textual words is more appropriate.

In this paper, we propose a deep matching process for TBMIR that is different from traditional information retrieval and NLP described as follows: first, it takes into account the specificity of images, by mapping the textual queries and the image metadata (document) into MDF. Second, it extracts the semantic relations between MDF, using UMLS, to build a good representation of query and document, and third, it computes the document relevance to the query by using the extracted relations.

In the literature, a variety of deep matching models have been proposed; however, most of them are designed for NLP, rather than information retrieval. Indeed, they consider the semantic matching instead of relevance matching. These models can be categorized, according to their architecture, into two types [14]: the first one is the interaction-focused models [15,16]. These models extract the relationships between queries and documents and then integrate them into a deep neural network to create new matching models. The second one is the representation-focused models [15,17]. These models apply the deep neural network to extract the best representations for both query and document, and then integrate them into a matching process.

In this paper, we propose a new medical image re-ranking process based on a deep matching model (DMM) for TBMIR. Overall, our model is a new representation-focused model that builds a good representation of queries and documents using MDF, UMLS and a personalized CNN for relevance matching. Specifically, we first create the semantic similarity matrix by extracting the UMLS relationships between each pair of MDF. Each query/document MDF is mapped into a similarity vector representing the relationships between the corresponding MDF and all MDF. As each query/document is composed of MDF, the resulting representation of the query/document will be a similarity matrix. Using these matrixes, our model tries to find the best representations for both query and document. Indeed, It applies a personalized CNN which is composed of retrieval filters taking into account some ranking features. Finally, an overall matching score is computed.

We evaluate the effectiveness of the proposed DMM using the ImageCLEF datasets from 2009 to 2012. For comparison, we take into account three well-known traditional retrieval models. The empirical results show that our model significantly outperforms the baseline models in terms of all the evaluation metrics.

This paper is structured as follows: Section 2 summarizes related work. Section 3 describes the proposed DMM model. Section 4 describes, first, how to represent MDF as a semantic similarity matrix using UMLS similarity, and second, our personalized CNN model with specific filters and finally the matching function. Experiments and results are presented and discussed in Section 7. Finally, Section 8 concludes the paper and gives some future work.

## 2. Related Work

Text-based indexing opens the possibility of indexing medical images by utilizing the associated reports, thereby providing a crucial way to access the exponentially growing clinical image databases. Current methods to retrieve images from medical databases are either based on attribute image content or non-image content. The content-based image retrieval (CBIR) method relies on image features such as shape, texture, or (in most successful cases) a previously assigned semantic feature to retrieve similar images. Non-image content retrieval methods often deal with text-based searches of databases, where a search query is submitted to an image database and text associated with images is compared to the query. However, both methods do not consider the actual medical knowledge relevant to an image and more traditional methods of organizing medical records which are indexed by keywords. Text-based image retrieval by and large builds upon retrieval technique that aims at finding all images relevant to a given query out of large database of images. In most textbook studies in proposed system, a specific image is given as an entry and the user wants to retrieve all images relevant to the given query. Though it is actually a subtype of text-based image retrieval where actual text-based images are not available and it is limited by the availability of some keyword-based annotation of the image. But this method has been shown to be very effective in retrieving images relevant to a given query and hence can be generalized to retrieval systems where text is the main modality. The creation of ’indexes of associated features’ from images is effectively the creation of a searchable database that links images to text.

In the literature, several works studied the use of CNN and semantics in medical image retrieval. This section briefly summarizes some of these works.

### 2.1. CNN for Medical Image Retrieval

The use of CNN models in the medical image retrieval domain has received great attention [18,19,20]. Authors in [21] used the CNN model based on the bag-of-word (BoW) technique to index biomedical articles. In this particular model, the input is a matrix of numbers that stand for the various medical terms that are contained in the input text. After that, a system of hidden layers is utilized to assign categories to the document. The authors of [22] developed yet another method for the classification of medical texts that may be put to use for retrieval operations. It does this by employing CNN training to extract the semantics of an input sentence; more specifically, it uses the Word2vec technique to represent the input sentences. This method is based on the use of CNNs. During the training of the CNN model, which is comprised of numerous hidden layers, it additionally maintains the list of stop-words. The CNN model was employed by the authors in the cited article [23] to remove the background noise from clinical notes that were going to be used for medical literature retrieval. They represented the input questions by using GloVe vectors, which are cited in the following reference: [24]. The CNN model’s primary purpose in this study is to make predictions about the relative relevance of search query phrases.

Despite the success of CNN for computer vision and NLP, employing CNN to search for relevant documents in TBMIR is not effective; and this may be due to the complexity of the ranking process. Moreover, most existing CNN models represent queries and documents without taking into account the specificity of the medical domain. This latter requires semantic extraction using external medical resources.

### 2.2. Semantics in Medical Image Retrieval

The integration of semantic knowledge in the medical image retrieval domain has received great attention, such as [25,26,27]. Authors in [28] used UMLS meta-thesaurus in the medical domain to improve queries and converting words to medical terms. They integrated the semantics in the retrieval process to map the text into concepts using UMLS meta-thesaurus [29]. The authors of the cited paper [30] developed a retrieval method to discover discriminative qualities between various medical photos using a Pruned Dictionary that was based on a description of a Latent Semantic Topic. They did this by calculating the topic-word relevance, which allowed them to make a prediction about the word’s relationship to the underlying topic. The latent themes are learned based on the association between the images and the words, and they are used to bridge the gap between low-level visual features and high-level semantic characteristics. This is accomplished by bridging the gap between low-level visual features and high-level semantic features. Moreover, in [31], an image retrieval framework that is based on semantic features has been proposed by the authors. This framework relies on (1) the automatic prediction of ontological terms that define the image content and (2) the retrieval of similar images by analyzing the similarity between annotations. The study of this system demonstrated that it is beneficial to make use of ontology while retrieving medical images.

Despite the large number of works using CNN and semantic resources in medical image retrieval, there is a lack of studies that investigate the integration of semantic knowledge on the CNN model to enhance the medical image retrieval performance. Therefore, we propose a new deep matching model based on personalized CNN and semantic resources (MDF, UMLS) to improve retrieval accuracy.

## 3. Overview of Our Approach

It is well known that medical images and their associated reports are not usually in agreement. For example, a patient who has a slipped disc, but displays no symptoms, will not have many abnormalities in his MR scan but will have many associated words or phrases about his condition. This inconsistency is a major problem for medical image retrieval systems, which rely on the images and associated text being “about the same thing”. With this in mind, and the fact that we have indexed the images and text separately, we need to devise a way where a text query can be used to aid the image query and vice-versa, without the user having to switch between the two.

Our proposed solution leverages relevance feedback to integrate information from both image and text queries using the other modality. For instance, if a user seeks to locate an image corresponding to a report about ‘left lung cancer’, the current system requires them to separately index the text using a natural language processing (NLP) tool and formulate a query, then repeat the process for the image. This method is inefficient and requires users to switch between modalities. In contrast, our system enables users to use an NLP tool to index the text or query, subsequently identifying relevant images that correspond to the text. Modality-specific technology subsequently ranks the images based on their similarity to the text. This approach automates the task of ’finding images matching this report,’ enhancing efficiency and accuracy.

Due to the positive impact of MDF on both retrieval performance [6,13] and query classification [32,33], we choose to integrate them into a deep matching model. In this study, we utilized the Unified Medical Language System (UMLS) as our semantic resource to construct a semantic similarity matrix, which represents the relationships between pairs of Medical Dependent Features (MDF). The literature [34,35,36] widely recognizes UMLS as a comprehensive thesaurus and ontology of biomedical concepts, designed to link various biomedical terminologies. By leveraging UMLS, we ensure a robust semantic framework for our analysis. Additionally, our system allows users to index text or queries using natural language processing (NLP) tools, facilitating more accurate and efficient retrieval of relevant medical information.

A new personalized CNN model using MDFs is proposed to build the best representation for both queries and documents, that are used to compute their matching score. In this paper, Figure 1 presents an overview of our approach:The preliminary step:

We represent each query/document as a set of features, then, for each MDF ($F_{i}$), we assign the corresponding vector extracted from the similarity matrix. Hence, each query/document is represented as a Matrix.

The Deep Matching Model Process:

**First**, we build a good representation of the query/document with a personalized CNN model that takes into account the interaction between query and document. Indeed, several personalized filters have been proposed and integrated into this model. Then, a matching function is applied to measure the matching degree between the query and the document representations. More precisely, we use the cosine similarity function as a matching function.

**Second**, we combine the obtained score with the corresponding baseline score to form a new re-ranking score. The re-ranking process is achieved by sorting the images according to their new scores.

## 4. Deep Matching Model: Preliminary Step

### 4.1. Medical Dependent Features

As our work falls into the medical image retrieval field, we propose to integrate the MDFs [6,32], that are a set of categorical medical features, into a new deep matching model to enhance the retrieval performance. A medical dependent features presented in the Figure 2.

Each MDF $f_{i}$ has *m* associated values *v* defined by $f_{i}=\left\{v1,v2,\ldots vm\right\}$. The set of MDF used in our work is detailed as follows:**Radiology** = “Ultrasound Imaging”, “Magnetic Resonance Imaging”, “Computerized Tomography”, “X-Ray”, “2D Radiography”, “Angiography”, “PET”, “Combined modalities in one image”, “Coronarography”, “Cystography”, “Scintigraphy”, “Mammography”, “Bone Densitometry”, “Radiotherapy”, “Urography”, “Pelvic Ultrasound”, “Myelography”, “FibroScan”**Microscopy** = “Light Microscopy”, “Electron Microscopy”, “Transmission Microscopy”, “Fluorescence Microscopy”, “Biopsy”, “Stool Microscopy”, “Capillaroscopy”, “Trophoblast Biopsy”, “Cytology”**Visible light photography** = “Dermatology”, “Skin”, “Endoscopy”, “Other organs”, “Colposcopy”, “Cystoscopy”, “Hysteroscopy”**Printed signals and waves** = “Electroencephalography”, “Electrocardiography”, “Electromyography”, “Holter”, “Audiometry”, “Urodynamic Assessment”**Generic Biomedical Illustrations** = “modality tables and forms”, “program listing”, “statistical figures”, “graphs”, “charts”, “screen shots”, “flowcharts”, “system overviews”, “gene sequence”, “chromatography”, “gel”, “chemical structure”, “mathematics formula”, “non-clinical photos”, “hand-drawn sketches”**Dimensionality** = “macro”, “micro”, “small”, “gross”, “combined dimensionality”**V-Spec** = “brown”, “black”, “white”, “red”, “gray”, “green”, “yellow”, “blue”, “colored”**T-spec** = “finding”, “pathology”, “differential diagnosis”, “Amniocentesis”, “Hemogram”, “Non-Invasive Prenatal Screening”, “Urinalysis”, “Lumbar Puncture”, “Seminogram”, “Triple Test”**C-spec** = “Histology”, “Fracture”, “Cancer”, “Benign”, “Malignant”, “Tumor”, “Pregnancy”, “Antibiogramme”

### 4.2. Semantic Matrix Construction

In this section, we present a new semantic mapping method using two semantic resources: MDF and UMLS. Frequently in NLP, the text data is converted into a vector of numbers, which deep models can process as input. Several approaches, such as Word2Vec, Glove, and one-hot-encoding, have been proposed for word embedding. Usually, these models consider general terminologies, which are not specific for any domain, to derive similarities and relations between words. As our work fits in the medical image retrieval field, we believe that retrieval performance could be improved if we use medical semantic resources such as UMLS for converting textual words. We represent the queries and documents as a set of MDF to keep only semantic information related to the medical domain. Then, each MDF is transformed into a concept using the MetaMap tool, then the UMLS Similarity tool [37] is used to calculate the similarities between each pair of concepts and then construct the similarity matrix as shown in Figure 1.

As shown in the preliminary step of Figure 1, all features are transformed into a similarity matrix and thus by following the next steps:Step 1: the MetaMap tool [38] is used to transform each MDF into a concept.Step 2: the similarities between each pair of medical concepts are calculated using the UMLS Similarity tool [37,39]. These semantic similarity scores are arranged in a semantic matrix. More precisely, we use the Resnik measure to determine the semantic relations between extracted concepts, as according to [40], it performs better than Path-based measures.

## 5. Deep Matching Model Construction

The new DMM model is a representation focused model that should build a good representation for a query and document with a deep neural network and conduct matching between the corresponding representations. Moreover, this model should take into account the specificity of information retrieval, NLP and medical image retrieval.

The inputs to our DMM model are a set of queries and documents presented with MDFs; each MDF is transformed into concepts then to a vector of numbers to be processed by the subsequent layers of the network. In the following, we detail the main components of our DMM model: the query/document matrix extraction, the personalized CNN and the matching function.

### 5.1. Query and Document Matrix Extraction

As our work fits in the medical image retrieval field, we represent the queries and the documents as a set of MDF in order to keep only semantic information related to the medical domain. In this paper, we propose to convert each query and document into an MDF vector. Then, each vector is converted to a semantic similarity matrix as presented in Figure 3:Step 1: For each query/document vector, we assign a binary value for each MDF depending on whether the query/document contains the feature value or not. The length of the resulting vector *V* equals *n* where *n* is the number of MDFs. This vector is transformed into a n∗n matrix *M*/∀i∈n,∀j∈n,M[i][j]=V[i] where *i* represents the row index and *j* represents the column index.Step 2: we multiply the resulting matrix M with the semantic similarity matrix SSM to obtain a new query matrix NQM as follows:NQM[i][j]=M[i][j]∗SSM[i][j]The illustration of the calculation is done in Figure 3.

### 5.2. Personalized CNN

We present, in this section, the personalized CNN that explicitly addresses the three specificities mentioned above. Indeed, the filters are designed to extract the best representation of queries and documents. In each representation, the network considers several retrieval features such as the MDF co-occurrence, the document ranking, and the IPM score. Moreover, it considers the NLP features for each query/document representation as it extracts the interaction between document and query. Indeed, according to [14], most of NLP models extract the interaction between two texts.

In the following, we present the layers of our network: convolutional, activations, pooling and fully connected layers.

Figure 4 Presents the architecture of the personalized CNN model.

#### 5.2.1. Convolutional Layer

In this layer, a set of filters $F\in R^{d}$ are applied to the query and document vectors to produce different feature maps. In our model, the query filters are distinct from the document filters. Below, we provide detailed information on the filters used for each component (document and query).

Query Filters:

The query filters aim to extract the best representation of the queries by considering the relationship between the document and the query. The more relevant the document is to the query, the higher the resulting vector values will be.

Confidence Query Filter (CoQF): The idea consists of calculating the co-occurrences of query MDFs with all MDFs.
(1)$$CoQF=\frac{\sum_{j\in Q}\sum_{i\in D}fr\left(f_{i},f_{j}\right)}{\sum_{i\in D}fr\left(f_{i}\right)}$$
where *Q* is the query MDF, *D* is the document MDFs, $fr(f_{j})$ is the cooccurrence of query MDFs in the collection, $fr(f_{i})$ is the cooccurrence of document MDFs in the collection and $fr(f_{i},f_{j})$ is the cooccurrence of query MDF and document MDF in the collection.In order to take into consideration, the length of the document, we use this filter. A document having only the query MDF should be more relevant than a document having other MDF in addition to the query ones. In fact, both documents are specific but the first document is more exhaustive. For that, we propose to divide the number of MDF in both document and query, with the number of document MDF. If the document did not include any query MDF, then the value will 0.Length Query Filter (LQF): For each query, if the document contains all query MDF, then we divide the number of MDF in both document and query, with the number of document MDF. Else, the value will be equal 0.
(2)$$LQF=\frac{\left|MDF\in (Q,D)\right|}{\left|MDF\in D\right|}$$
where $\left|MDF\in (Q,D)\right|$ is the number of MDFs in both query MDF *Q* and document MDF *D* and $\left|MDF\in D\right|$ is the number of MDF in the document containing all query MDF.Rank Query Filter (RQF): We calculate the inverse document rank. If the document did not appear in the first search, the RQF will be equal.
(3)$$RQF=\frac{1}{docrank}$$Proximity Query Filter (PQF): IIn the event that a document has query MDFs, we will compute the inverse of the distances that separate these MDFs in the document. In this instance, the distance between two features is represented by the total number of features that are located between them.
(4)$$PQF=\frac{1}{1+\sum distMDF\in D}$$
where distMDF∈D is the distances between document MDFs.PMI Query Filter (PMIQF): The PMI (Pointwise Mutual Information) [41] is a proposed metric to find features with a close meaning. Indeed, the PMI of the MDFs $f_{i}$ and $f_{j}$ is defined using the occurrences of $f_{i}$ ($fr(f_{i})$) and $f_{j}$ ($fr(f_{j})$), the co-occurrences $fr(f_{i},f_{j})$ within a vector of features, and *N* is the collection size.
(5)$$PMIF\left(QF\right)=log\frac{N\times fr(f_{i},f_{j})}{fr\left(f_{i}\right)\times fr\left(f_{j}\right)}$$This equation calculates the semantically closest MDFs of the collection to $f_{i}$ and $f_{j}$.Feature Difference Query Filter (FDQF): The more the query MDFs not found is small, the more the document is relevant. For each query, we compute the inverse of number of query MDFs not in document MDFs.
(6)$$FDQF=\frac{1}{\sqrt{1+\left|MDF\in \left\{Q\cap D\right\}\right|}}$$

Document Filters:

Similar to the query filters, document filters try to extract the best representation of documents. They are based on the relationship between document and query. The more the document is relevant to the query, the highest is the resulting vector values.

Confidence Document Filter (CoDF): This document filter determines the total amount of MDF documents that are included in the query. The relevance of the document will increase in proportion to the number of query MDFs it contains.
(7)$$CoDF=\sum \left(f_{i}\underline{q}\cap f_{j}\underline{d}\right)$$
where $\left(f_{i}\underline{q}\cap f_{j}\underline{d}\right)$ is the number of common MDF in query.Length Document Filter (LDF): When it comes to documents, first we determine the number of document MDFs that are included in the related query, and after we have that amount, we divide it by the document length (LD). In point of fact, the relevance of the document will increase if it is of a modest size and if it shares several characteristics with the query being conducted.
(8)$$LDF=\frac{\left|MDFdocinquery\right|}{LD}$$
where $\left|MDFdocinquery\right|$ is the number of MDF in both document and query and LD is the document length using the MDF features.Rank Document Filter (RDF):
(9)$$RDF=\sum_{i\in q}fr\left(f_{i}indoc\right)\times \gamma$$The variable $fr(f_{i})$ represents the frequency of query MDFs in the document, while γ represents the organization factor of the query in the document. The value of γ is 1 if the query preserves its organization in the document, and 0.5 if it does not.Proximity Document Filter (PDF): The more the document’s features existing in the query are closer, the more it is relevant.
(10)$$PDF=\frac{1}{\left|f_{i}\in Q\right|}$$
where FD∈Q is the documents MDFs in the query.PMI Document Filter (PMIDF): Similar to PMI in query filter, PMI in document filter try to find MDFs with a close meaning. It has the same equation except the *N* in this filter is the document size.
(11)$$PMIF\left(DF\right)=log\frac{N\times fr(f_{i},f_{j})}{fr\left(f_{i}\right)\times fr\left(f_{j}\right)}$$This equation calculates the semantically closest MDFs in the document.Feature Difference Document Filter (FDDF): The more the number of document MDFs not in the query is small, the more the document is relevant.
(12)$$FDDF=\frac{1}{\sqrt{1+\left|MDF\in D-Q\right|}}$$
where *D* is the document MDFs and *Q* is the query MDFs.

The input of the DMM model is a matrix $S\in R^{n\times n}$, and the convolutional filters are also matrices $F\in R^{n}$. It is important to note that these filters have the same dimensionality, denoted as *n*, as the input matrix. In addition, these filters scan the vector representations and produce an output vector $C\in R^{m}$. Each component $c_{i}$ of the vector *C* is obtained by multiplying a vector *V* with a filter *F*, and then summing the resulting values to obtain a single value.
(13)$$c_{i}=\sum_{k=1}^{n}V_{k}F_{k}$$

#### 5.2.2. Activation Function

Immediately after the convolutional layer comes a non-linear activation function called alpha that is applied to the output of the layer that came before it. Through the use of this function, it is possible for a neuron’s input signal to be transformed into an output signal. In the research that has been done, a number of different activation functions have been proposed [42]. One of these functions is called the Rectified Linear Unit (ReLU) function, and it assures that positive values are passed on to the subsequent layer. The authors in [43] demonstrated that this function is effective, uncomplicated, and has the capacity to lower the amount of complexity as well as the amount of time required for calculations. As a result, we have decided to include this function in our model in the capacity of an activation function.

#### 5.2.3. Pooling Layer

The pooling layer’s goal is to do three things: aggregate information, minimize the amount of representation used, and derive global features from the convolutional layer’s local ones. There are two functions that can be found in the body of literary work: (1) the average consists of computing the average of each feature map of the convolutional layer to consider all the elements of the input are even if many of them have low weights [44], and (2) the Max consists of selecting the maximum value of each feature map of the convolutional layer. Both of these operations are performed in order to take into consideration all of the elements of the input are. We have decided to adopt max-pooling for our research because it takes into account only neurons with high activation values, which ultimately results in a high level of semantic abstraction of the input data.

#### 5.2.4. Fully Connected Layer

In order to produce a final vector representation of the query or document, a Fully Connected Layer (FCL) is applied to the vector that was generated as a result of the previous step.

### 5.3. Matching Function

According to [14], the most significant challenge associated with the retrieval of information is the matching problem, which refers to the process of determining a document’s relevancy in light of a query. If we have a document denoted by *d* and a query denoted by *q*, then the matching function is a mechanism for assigning a score to the representation of *d* and *q*:(14)RSV(d,q)=F(Φ(d),Φ(q))
where *F* stands for the scoring function and Phi is the mapping function that converts each *d*|*q* pair into a vector representation. In the research that has been done on the subject, a number of different deep matching models have been suggested for the overall matching process. These silhouettes fall primarily into one of two categories when grouped together. The representation-focused model is the first one, and in this model, Phi is a complicated mapping function while *F* is a straightforward scoring function. A deep neural network is utilized by this model in order to construct an accurate representation for the document as well as the query. After that, it does some sort of matching between these different representations. The second one is the interaction-focused model where Φ is simple mapping function and *F* is complex scoring function.

We use a representation-focused model in which Phi is a sophisticated mapping function between representations and *F* is a straightforward matching function. Since the sophisticated Phi-based mapping function of the individualized CNN is what drives our selection, we resort to the more elementary *F*-based cosine similarity. The formal definition of a document’s relevance to a query is as follows:(15)$$RSV\left(Q,D\right)=cosine\left(\vec{Q},\vec{D}\right)=\frac{\vec{Q}.\vec{D}}{∥\vec{Q}∥.∥\vec{D}∥}$$
where $\vec{Q}$ and $\vec{D}$ are the query and the document vectors respectively. In the IR, for a given query, the documents are ranked by their relevance scores.

## 6. SemRank: Semantic Re-Ranking Model Based on DMM

In the last part of this article, we discussed our MDF-based deep matching model, which calculates the DMM score of the document *d* with respect to the query *q*. However, doing a search of relevant documents by utilizing MDF alone is insufficient; certain phrases could not be mapped to MDF, and as a result, such keywords should be eliminated from the search. As a result, we recommend combining the findings of the DMM with those of the baseline, taking into consideration all of the query terms. To be more specific, we suggest modeling the SemRank score using the most common type of late fusion approach, which is known as a straightforward linear combination. Before adding the two scores together, we first standardize the initial score and the DMM score in the following manner:(16)$$SemRank_{score}=\alpha *\frac{InitialScore}{\max InitialScores}+\left(1-\alpha \right)*\frac{DMMScore}{\max DMMScore}$$
where α is a balancing parameter α∈[0…1], InitialScore represents the initial ranking score of the document and DMM score of the same document. The normalized score is obtained by dividing the relevance score for a given document *d* by the highest relevance score in the whole collection. As a baseline, we propose to use the BM25 model which is well known for its efficiency and its performance in many retrieval tasks

## 7. Experiments and Results

In this section, we describe the experimental datasets, then we present our several experiments released to evaluate the accuracy of our model and we compare it to some existing approaches.

### 7.1. Experimental Datasets

In order to assess the effectiveness of our suggested method, it is imperative to utilize medical image datasets that include both images and textual descriptions, together with queries and ground truth. The majority of medical data sets currently available do not fulfill these criteria. Some sources lack assessment protocols, such as OHSUMED [45], while others focus on textual analysis and evaluation, like TREC. On the other hand, the ImageCLEFmed evaluation campaign offers specific medical picture collections for the purpose of assessing medical image retrieval. From 2011 onwards, the quantity and extent of the collections were comparable to those seen in real-world applications [46]. Due to copyright restrictions, the redistribution of the ImageCLEFmed collections to research groups is only allowed through a special agreement with the original copyright holders [47]. Therefore, we are restricted to conducting experiments using only the five collections for which we have obtained copyrights. The collections are shown in Table 1 and consist of two relatively small data sets: 74,902 and 77,495 images for the 2009 [48] and 2010 [49] data sets, respectively. After the evolution of ImageCLEF, three additional data sets were added: 230,088 images for the 2011 [50] data set, and 306,539 images for both the 2012 [51] and 2013 [52] data sets.

Each image in these data sets is accompanied by a textual description. An image can have the text from its caption or a hyperlink to the HTML page that has the complete text of the article [53], together with the title of the article [48]. Furthermore, the queries were chosen from a collection of themes suggested by physicians and clinicians in order to precisely mimic the information requirements of a clinician involved in diagnostic tasks. The images in the 2009 and 2010 data sets were sourced from the RSNA journals Radiology and Radiographics [48] and consist of a portion of the Goldminer data set. Nevertheless, the photos in the data sets from 2011, 2012, and 2013 are derived from studies that were published in open-access journals and may be accessed through PubMed Central. The later data sets have a wider range of images, including charts, graphs, and other nonclinical images, resulting in more visual variety.

To evaluate the proposed SemRank model, we conducted experiments using the ImageCLEF collections in Medical Retrieval Task from 2009 to 2012. These collections are composed of images and queries. Each image has a textual description. Each query is composed of text representation and a few sample images. In our work, we use only text representation of the queries and textual description of the images. The 2009 and the 2010 datasets are relatively small (74,902 and 77,495 images respectively). The 2011 and the 2012 datasets are significantly bigger (230,088 and 306,539 images respectively). Indeed, these datasets contain a greater image diversity and also include charts, graphs and other non-clinical images [32]. Also, we are limited to these datasets as we do not have the 2013 dataset, the last dataset in the medical image retrieval task of imageCLEF. Figure 5 illustrates 3 images of the used datasets, respectively with their associated MDF, extracted using the Medical-Dependent Features. We observe here that MDF features represent specific characteristics of medical images but not a body part (brain) or a pathology (cancer) and this due to the nature of medical textual queries aiming to find medical images.

### 7.2. Effectiveness of the SemRank Model in Image Reranking

In this section, we present a set of experiments carried out to the SemRank model. To achieve the best linear combination, we use several values of α. α = 0 means that only the DMM score is used and α = 1 means that only the BM25 score is used. Figure 6 presents the MAP, the P@5 and the P@10 values when: α∈[0:1] in datasets from 2009 to 2012.

According to Figure 6, we notice that using only DMM model to retrieve relevant documents gives the worst ranking results. However, the combination of the baseline and the DMM models gives better results. This proves our assumption that using only MDF to search for relevant documents is not sufficient; the combining models is a solution. According to MAP, P@5 and P@10 values, the best results are obtained when α∈[0.1:0.5]. In the remaining experiments, we chose to set α = 0.3.

### 7.3. Comparison of the SemRank Model with Literature Models

In this section, we propose to compare our model with BM25, DLM (Dirichlet Language Model) [54] and Bo1PRF (Bo1 Pseudo Relevance Feedback) [55] models. Table 1 summarizes this comparison according to the P@5, P@10 and MAP measures. The best result of all models and for each metric is presented in bold.

For the 2009 and the 2010 ImageClef datasets, the results show that our SemRank model performs better than the existing models in terms of MAP, P@5 and P@10. For the 2011 dataset, the SemRank model gives better results than the BM25 and the DLM models in terms of P@5 and P@10, but do not outperform the Bo1PRF model in term of MAP. Moreover, For the 2012 dataset, the Bo1PRF model outperforms our model in terms of MAP, P@5 and P@10. This can be explained by the high number of non-clinical images in these datasets which contain a diversity of images (tables, shapes, graphs); and our retrieval model is specific for medical images. Moreover, the Bo1PRF model is based on the pseudo-relevance feedback technique that improves retrieval results.

The accuracy gain is presented in Table 2. Indeed, we determine the improvement rate and we conduct a statistical *t*-test (Wilcoxon) [56] to evaluate the results. The gain is considered statistically significant when p<0.05. In this work, the results are followed by the ** when p<0.05.

Results show that the improvements have been achieved on the majority of datasets.Our model achieves between 12% and 29% on the 2009 dataset, which is a substantial improvement over the performance of existing models. When compared to the DLM and Bo1PRF models, the retrieval performance of the 2010 dataset is significantly enhanced by the SemRank model. This might be explained by the fact that the datasets from 2009 and 2010 contain photos suggested by clinicians and physicians that beat the information that is required.

Although our model is performing worse than the Bo1PR model on the 2011 and the 2012 datasets, it improves the retrieval performance (+4 percentage points) compared to the BM25 model for the 2012 dataset and (+40 percentage points) compared to the DLM model for the 2011 dataset. This variation may be due to the pseudo relevance feedback technique, which adds the first m keywords that appear in the top k retrieved documents. However, our model uses only query features without additional terms and enhances significantly the retrieval performance on the 2009 and the 2010 datasets.

We conclude that using our DMM improves significantly the results compared to the literature models. This validates our assumption that our DMM is a promising technique for improving medical image retrieval performance. In addition, this improvement could be related to the importance of using medical external resources: MDF and UMLS.

## 8. Conclusions and Future Work

This paper introduces an innovative SemRank model designed to enhance the ranking of medical images. The model leverages two external semantic resources: the Medical-Dependent Features (MDF) terminology and the Unified Medical Language System (UMLS) Metathesaurus. Within this framework, queries and documents are represented as sets of MDF, with the UMLS ontology employed to compute semantic similarity matrices between these sets. These matrices serve as the foundation for constructing matrix representations for each query and document, which are subsequently integrated into a Convolutional Neural Network (CNN) process. The resulting outputs yield vectors used to compute new relevance scores for documents when presented with a query. This innovative approach not only harnesses semantic knowledge from external resources but also employs advanced neural network techniques to improve the accuracy and effectiveness of medical image retrieval.

Our experiments were conducted on the Medical ImageCLEF collections from 2009 to 2012. The findings demonstrate a significant improvement in the re-ranking process when integrating Medical-Dependent Features (MDF) and the Unified Medical Language System (UMLS) into the Deep Matching Model (DMM). Furthermore, a comparative analysis was conducted between our model and various state-of-the-art approaches. The results revealed a noteworthy increase in the accuracy of the re-ranking process, underscoring the efficacy of our proposed methodology.

In our forthcoming research endeavors, we aim to augment the capabilities of the CNN model by integrating supplementary filters that encompass a broader spectrum of retrieval attributes. Furthermore, we plan to enhance the SemRank model by incorporating visual features, thereby elevating the precision of image retrieval.

## Figures and Tables

**Figure 1 diagnostics-14-01204-f001:**
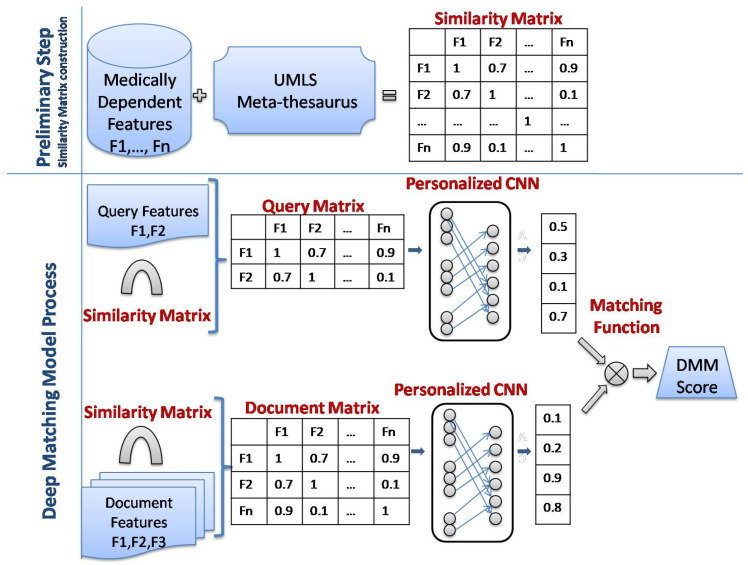
Overview of the Deep Matching Model.

**Figure 2 diagnostics-14-01204-f002:**
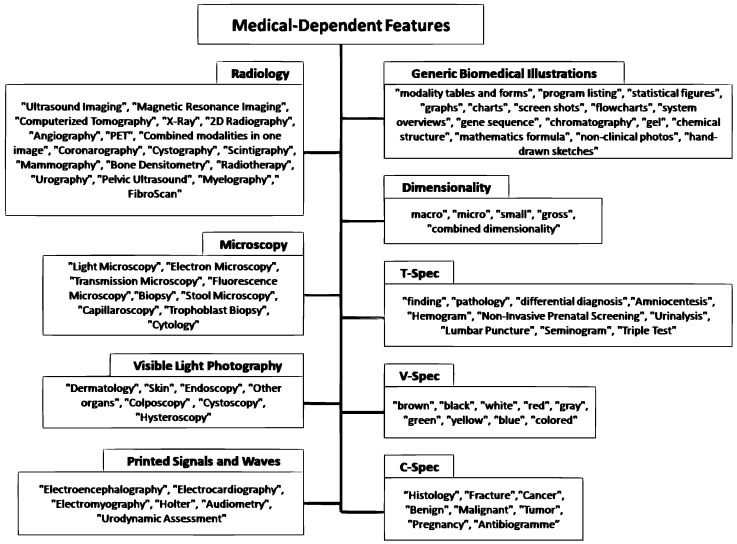
Medical -Dependent Features.

**Figure 3 diagnostics-14-01204-f003:**
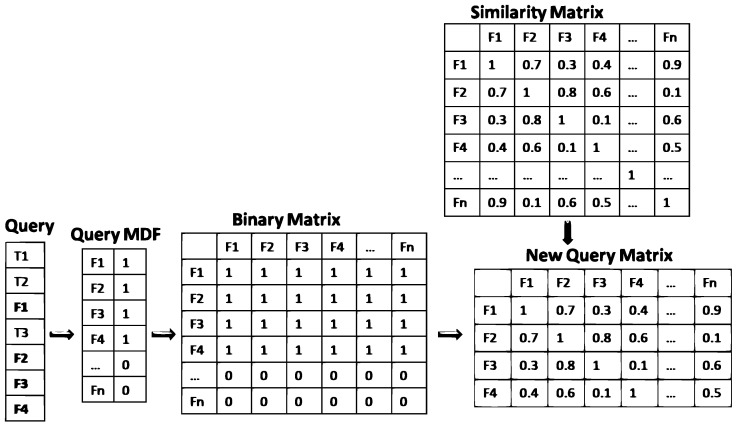
Query matrix extraction process.

**Figure 4 diagnostics-14-01204-f004:**
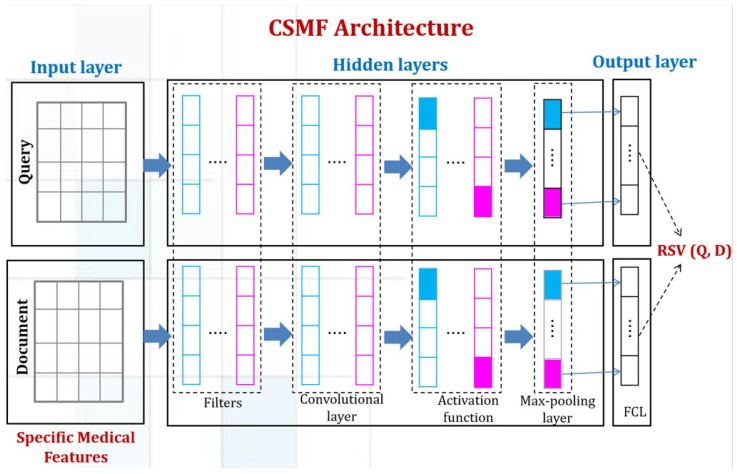
The architecture of the personalized CNN model.

**Figure 5 diagnostics-14-01204-f005:**
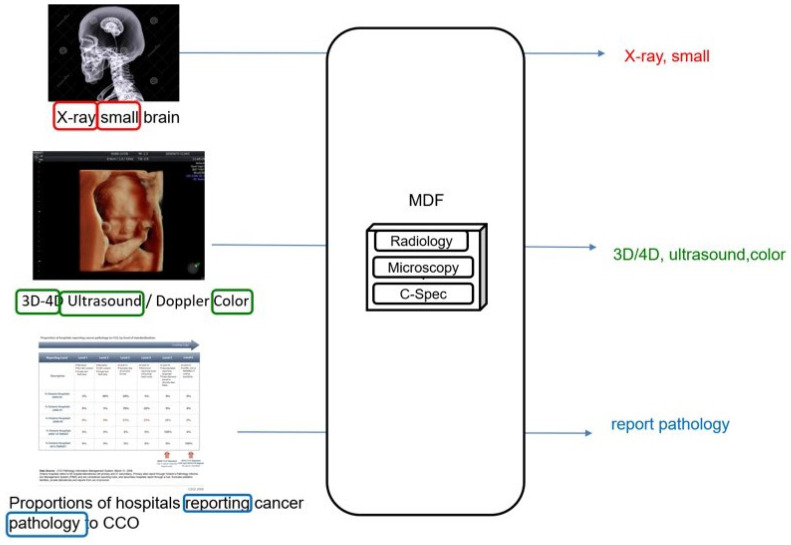
Some examples of ImageCLEF medical images and their extracted MDF.

**Figure 6 diagnostics-14-01204-f006:**
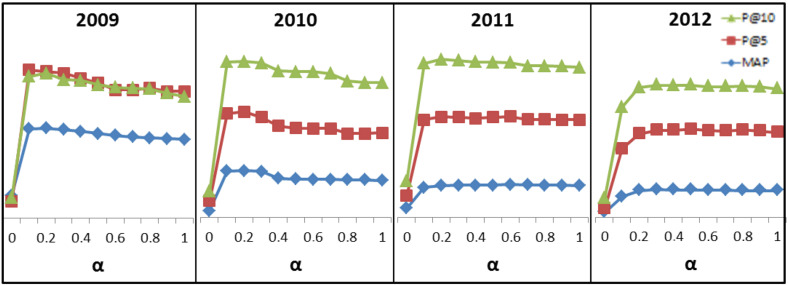
Results according to α using 4 ImageCLEF datasets 2009, 2010, 2011 and 2012.

**Table 1 diagnostics-14-01204-t001:** Comparative results with the previous state-of-the-art approaches using ImageCLEF datasets.

		BM25	DLM	Bo1PRF	SemRank (α = 0.3)
**ImageClef-2009**	**P@5**	0.608	0.592	0.608	**0.696**
	**P@10**	0.584	0.524	0.568	**0.664**
	**MAP**	0.379	0.327	0.371	**0.425**
**ImageClef-2010**	**P@5**	0.400	0.436	0.361	**0.453**
	**P@10**	0.420	0.375	0.330	**0.453**
	**MAP**	0.312	0.313	0.305	**0.389**
**ImageClef-2011**	**P@5**	0.393	0.240	0.386	**0.406**
	**P@10**	0.313	0.223	0.326	**0.340**
	**MAP**	0.193	0.138	**0.211**	0.195
**ImageClef-2012**	**P@5**	0.418	0.281	**0.554**	0.427
	**P@10**	0.313	0.241	**0.409**	0.322
	**MAP**	0.193	0.146	**0.361**	0.201

**Table 2 diagnostics-14-01204-t002:** Accuracy gain of the SemRank compared to other models.

	2009	2010	2011	2012
**SemRank/BM25**	+12% **	+24%	+1%	+4%
**SemRank/DLM**	+29% **	+24%	+40% **	+38%
**SemRank/Bo1PRF**	+14% **	+27% **	-	-

## Data Availability

The data used in this study are openly accessible at the following link: https://www.imageclef.org.

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
