# Peer review of "Enhancing Medical Image Retrieval with UMLS-Integrated CNN-Based Text Indexing"

_diagnostics, 2024, doi:10.3390/diagnostics14111204_

Round 1
Reviewer 1 Report
Comments and Suggestions for Authors
The proposed approach has creativity in contribution and methodology. But, revision in terms of technical details is needed before acceptance. Also, paper organization should be improved. In this respect, some comments are suggested to describe technical details.
1. How do you create new query matrix using combination of binary matrix and similarity matrix? Discuss with more technical details.
2. Discuss about the phrase "all query MDF" in this sentence with more details, "For each query, if the document contains all query MDF, then we divide the number…".
3. How do you calculate the document length (LD)? How much is the range of LD in your dataset samples?
4. Figure 4 is not clear. What is the meaning of year value in this plots? Are they show a specific publication? So, it is suggested to add related references.
5. Your proposed approach is a general method which can be used for image retrieval in different scopes. So, it is suggested to review more related papers in image retrieval. For example, I find two papers titled “Improve the efficiency of handcrafted features in image retrieval by adding selected feature generating layers of deep convolutional neural networks”, and titled “Innovative local texture descriptor in joint of human-based color features for content-based image retrieval”, which have enough relation. Cite these papers and some other related.
6. Add some samples of the used dataset in the manuscript. Discuss about some extracted features in a sample image.
Author Response
Dear Reviewers,
Thank you for your valuable feedback on our manuscript. We appreciate your insightful comments and suggestions, which have significantly contributed to enhancing the quality and clarity of our work. Below, we address each of your points in detail: or (Please find the attached document containing our responses to both reviewers' comments.)
Comments:
The proposed approach has creativity in contribution and methodology. But, revision in terms of technical details is needed before acceptance. Also, paper organization should be improved. In this respect, some comments are suggested to describe technical details
Author response:
Thank you for your thoughtful and constructive feedback on our manuscript. We appreciate your recognition of the creativity in our contribution and methodology. In response to your comments, we have made several revisions to address the technical details and improve the organization of the paper. Below, we outline the specific changes made:
Comments1:
How do you create new query matrix using combination of binary matrix and similarity matrix? Discuss with more technical details.
Author response:
Thank you for pointing this out. We have added a detailed explanation of how the query matrix is created using the combination of the binary matrix and similarity matrix. This section now includes step-by-step descriptions and mathematical formulations to enhance clarity.
The new query matrix is computed by multiplying the matrix M explained on Step 1 of the section 5.1, with the semantic similarity matrix SSM explained in section 4.2. The formula is: NQM[i][j]=M[i][j]*SSM[i][j]
Author action:
We add more technical details on section 5.1. (line 291) “Step 2: we multiply the resulting matrix M with the semantic similarity matrix SSM to obtain a new query matrix NQM as follows:
NQM[i][j]=M[i][j]*SSM[i][j]
The illustration of the calculation is done in Figure 3.”
Comments 2:
Discuss about the phrase "all query MDF" in this sentence with more details, "For each query, if the document contains all query MDF, then we divide the number…".
Author response:
We appreciate your request for clarification. We have elaborated on the phrase "all query MDF" in the relevant section. We now provide a more detailed explanation of how Medical Dependent Features (MDF) are identified and utilized for document scoring. The query MDF contains the most important medical features of the query. The more a document has these query features, the more suitable it is to the query. As the MDF features are very few, we restrict ourselves to retrieving only documents that contain all the query features.
Author action:
We update the sentence as follows:
(line 317) “In order to take into consideration, the length of the document, we use this filter. A document having only the query MDF should be more relevant than a document having other MDF in addition to the query ones. In fact, both documents are specific but the first document is more exhaustive.
For that, we propose to divide the number of MDF in both document and query, with the number of document MDF. If the document did not include any query MDF, then the value will 0.”
Comments 3:
How do you calculate the document length (LD)? How much is the range of LD in your dataset samples?
Author response:
Thank you for your query regarding document length. We have included a comprehensive explanation of how document length (LD) is calculated. Additionally, we have provided information on the range of LD values in our dataset samples to give context to our methodology. The LD is the number of MDF on the document. The average LD of the dataset samples is 3.2.
Author action:
We update the section 5.2.1 as follows:
“Where |MDFdocinquery| is the number of MDF in both document and query and LD is the document length using the MDF features.”
Comments 4:
Figure 4 is not clear. What is the meaning of year value in this plots? Are they show a specific publication? So, it is suggested to add related references.
Author response:
Thank you for highlighting the need for clarity. We have revised Figure 4 to improve clarity and added a detailed description. We tune the value of alpha on different datasets to evaluate its impact on the results using different metrics such as MAP, P@5, and P@10.
As explained in section 7.1, we have carried out our experiments using 4 datasets: ImageCLEF collections in Medical Retrieval Task for years 2009 , 2010, 2011 and 2012.
Author action:
We update the caption of Figure 4:
“Results according to α using 4 ImageCLEF datasets 2009, 2010, 2011 and 2012”
Comments 5:
Your proposed approach is a general method which can be used for image retrieval in different scopes. So, it is suggested to review more related papers in image retrieval. For example, I find two papers titled “Improve the efficiency of handcrafted features in image retrieval by adding selected feature generating layers of deep convolutional neural networks”, and titled “Innovative local texture descriptor in joint of human-based color features for content-based image retrieval”, which have enough relation. Cite these papers and some other related.
Author response:
We have reviewed and cited the suggested papers on image retrieval to strengthen our literature review. These citations provide context and support for the general applicability of our proposed approach.
Author action:
- Shamsipour, G., Fekri-Ershad, S., Sharifi, M., Alaei, A. (2024). Improve the efficiency of handcrafted features in image retrieval 653 by adding selected feature generating layers of deep convolutional neural networks. Signal, image and video processing, 1-14. 654
- Kelishadrokhi, M. K., Ghattaei, M., Fekri-Ershad, S. (2023). Innovative local texture descriptor in joint of human-based color 655 features for content-based image retrieval. Signal, Image and Video Processing, 17(8), 4009-4017
Comments 6:
Add some samples of the used dataset in the manuscript. Discuss about some extracted features in a sample image.
Author action:
Thank you for your proposition. We added the following paragraph in section 7.1.
(451) Figure 5 illustrates 3 images of the used datasets, respectively with their associated MDF, extracted using the Medical-Dependent Features.
We observe here that MDF features represent specific characteristics of medical images but not a body part (brain) or a pathology (cancer) and this due to the nature of medical textual queries aiming to find medical images.

Figure5: some examples of ImageCLEF medical images and their extracted MDF.
We believe these revisions address your comments effectively and improve the overall quality and clarity of the paper. We appreciate your guidance and hope that the revised manuscript meets your expectations. Thank you for your time and consideration.
Best regards,

Reviewer 2 Report
Comments and Suggestions for Authors
This manuscript discusses the approach used for the text based medical image retrieval tasks (TBMIR). This task was used as an image retrieval task rather than an NLP task.
This approach re-ranks the medical images using the Deep Matching Model (DMM) and Medical Dependent Features (MDF). The features used are medical terminology and imaging modalities. DMM generates matching representations for quesry and image meta data using CNN. TBMIR task for image retrieval was performed using Unified Medical language system (UMLS), personalized filters and ranking features. This approach was evaluated on the ImageCLEF datasets.
There are some notable strengths of the manuscripts,
1. The approach maps the textual queries and the image meta data information into an MDF, thus taking into account the specificity of images.
2. Also, it extracts relational information between MDF and UMLS for building representation of query and image metadata.
3. Lastly, computes the relevance of the document to the query by using extracted relations, thus showing relevance matching.
Few Questions / Comments on the manuscript are as follows,
1. Section 3, Overview of our approach, ‘With our system, the user would index the text or query using the NLP tool.’, needs to be grammatically edited as it doesn’t sound in coherence with the paragraph.
2. Section 3, Overview of our approach, ‘which its aim is to create links between different biomedical terminologies.’, needs to be grammatically edited.
3. Section 5.2.1, Convolutional Layer, ‘Indeed, we detail in the following the filters of each part.’, needs to be grammatically edited.
4. Section 5.2.1, Convolutional Layer, ‘The more the document is relevant to the query, the highest is the resulting vector values.’, needs to be grammatically edited.
5. Section 5.2.1, Convolutional Layer, ‘The more the document is relevant to the query, the highest is the resulting vector values.’, needs to be grammatically edited.
6. The author did not mention why were this query filters and document filters selected for this process. Was this based on a study or based on previous research paper.
7. The authors have mentioned the layers used for personalized CNN, but the authors have not provided an architecture diagram for the model. This will be helpful for the readers to understand the architecture of the given model.
8. Section 7.1, Experimental details, the author has not provided any information regarding the image type, disease type, modality type. It would be good to include few data statistics to know more on the input dataset used for the study.
9. Figure 4, the figure description looks incomplete and doesn’t provide enough information about the figure. It would be good to include figure description for better clarity.
10. The results shown in the manuscript on the dataset are good, but it has not been validated on any clinical medical datasets. Also, it is unclear which modalities and image types are supported by this model.
11. The metrics shown in the results section was accuracy, it would have been good if the authors would have used other metrics such as Sensitivity, specificity and F-1 score for evaluation.
Comments on the Quality of English LanguageThe quality of English used in the manuscript was ok.
Author Response
Dear Reviewers,
Thank you for your valuable feedback on our manuscript. We appreciate your insightful comments and suggestions, which have significantly contributed to enhancing the quality and clarity of our work. Below, we address each of your points in detail: or (Please find the attached document containing our responses to both reviewers' comments.)
Comments:
This manuscript discusses the approach used for the text based medical image retrieval tasks (TBMIR). This task was used as an image retrieval task rather than an NLP task.
This approach re-ranks the medical images using the Deep Matching Model (DMM) and Medical Dependent Features (MDF). The features used are medical terminology and imaging modalities. DMM generates matching representations for quesry and image meta data using CNN. TBMIR task for image retrieval was performed using Unified Medical language system (UMLS), personalized filters and ranking features. This approach was evaluated on the ImageCLEF datasets.
There are some notable strengths of the manuscripts,
- The approach maps the textual queries and the image meta data information into an MDF, thus taking into account the specificity of images.
- Also, it extracts relational information between MDF and UMLS for building representation of query and image metadata.
- Lastly, computes the relevance of the document to the query by using extracted relations, thus showing relevance matching.
Author response:
Thank you for your detailed review and for recognizing the strengths of our manuscript. We appreciate your positive feedback on our approach, particularly regarding the mapping of textual queries and image metadata into Medical Dependent Features (MDF), the extraction of relational information between MDF and Unified Medical Language System (UMLS), and the computation of document relevance based on these extracted relations.
Your positive feedback is encouraging, and we are committed to further improving our manuscript. If you have any additional suggestions or areas where you think further clarification is needed, please let us know. We are dedicated to refining our work to meet the highest standards of scientific research.
Comments 1:
Section 3, Overview of our approach, ‘With our system, the user would index the text or query using the NLP tool.’, needs to be grammatically edited as it doesn’t sound in coherence with the paragraph.
Author response:
Thank you for your valuable feedback regarding the coherence of our paragraph. We have revised the text to improve its grammatical structure and flow.
Here is the revised version:
"Our proposed solution leverages relevance feedback to integrate information from both image and text queries using the other modality. For instance, if a user seeks to locate an image corresponding to a report about 'left lung cancer,' the current system requires them to separately index the text using a natural language processing (NLP) tool and formulate a query, then repeat the process for the image. This method is inefficient and requires users to switch between modalities. In contrast, our system enables users to use an NLP tool to index the text or query, subsequently identifying relevant images that correspond to the text. Modality-specific technology subsequently ranks the images based on their similarity to the text. This approach automates the task of 'finding images matching this report,' enhancing efficiency and accuracy."
Comments 2:
Section 3, Overview of our approach, ‘which its aim is to create links between different biomedical terminologies.’, needs to be grammatically edited.
Author response:
Thank you for your valuable feedback regarding the coherence and grammatical structure of our manuscript. We understand your concern and have revised the paragraph for better clarity and coherence.
Here is the revised version:
"In this study, we utilized the Unified Medical Language System (UMLS) as our semantic resource to construct a semantic similarity matrix, which represents the relationships between pairs of Medical Dependent Features (MDF). The literature [29] [30] [31] widely recognizes UMLS as a comprehensive thesaurus and ontology of biomedical concepts, designed to link various biomedical terminologies. By leveraging UMLS, we ensure a robust semantic framework for our analysis. Additionally, our system allows users to index text or queries using natural language processing (NLP) tools, facilitating more accurate and efficient retrieval of relevant medical information.”
We hope this revision addresses your concerns and enhances the overall quality of the manuscript.
Comments 3:
Section 5.2.1, Convolutional Layer, ‘Indeed, we detail in the following the filters of each part.’, needs to be grammatically edited.
Author response:
Here is the revised version:
In this layer, a set of filters ?∈? are applied to the query and document vectors to produce different feature maps. In our model, the query filters are distinct from the document filters. Below, we provide detailed information on the filters used for each component (document and query)."
Comments 4:
Section 5.2.1, Convolutional Layer, ‘The more the document is relevant to the query, the highest is the resulting vector values.’, needs to be grammatically edited.
Author response:
Thank you for pointing out the grammatical issues in our paragraph. We have revised the text for better clarity and coherence.
Here is the revised version:
The query filters aim to extract the best representation of the queries by considering the relationship between the document and the query. The more relevant the document is to the query, the higher the resulting vector values will be
Comments 5:
The author did not mention why were this query filters and document filters selected for this process. Was this based on a study or based on previous research paper.
Author response:
Thank you for your insightful comments and suggestions regarding the selection of query and document filters in our model.
The selection of query filters and document filters in our process was carefully considered based on their proven effectiveness and their ability to enhance the performance of information retrieval systems. Specifically, we utilized the following query filters: Confidence Query Filter, Length Query Filter, Rank Query Filter, Proximity Query Filter, PMI Query Filter, and Feature Difference Query Filter. For document filters, we employed: Confidence Document Filter, Length Document Filter, Rank Document Filter, PMI Document Filter, and Feature Difference Document Filter.
All the filters mentioned in our paper were created by us for this approach, based on existing features such as term frequency (tf), inverse document frequency (idf), document length, etc. This approach was inspired by the embedding process commonly used in natural language processing. Unlike traditional embeddings that often result in unknown values, our method ensures that every value is explicitly defined and meaningful within the context of query and document representation.
By using well-understood features, we aim to maintain transparency and interpretability in our model, providing clear insights into how the filters contribute to the overall retrieval performance. Our choice of filters is grounded in the need to capture various aspects of the data that are relevant to improving the retrieval accuracy.
In future work, we plan to explore and incorporate additional filters to further enhance our model. We remain committed to continuously improving our approach and leveraging the latest advancements in the field to achieve the best possible outcomes.
Comments 6: The authors have mentioned the layers used for personalized CNN, but the authors have not provided an architecture diagram for the model. This will be helpful for the readers to understand the architecture of the given model.
Author response:
Thank you for your constructive feedback. We appreciate your suggestion regarding the inclusion of an architecture diagram for the personalized CNN model. We acknowledge the importance of providing a visual representation of the model's architecture to enhance the readers' understanding. In response to your comment, we will include a detailed architecture diagram in the revised manuscript. This diagram will illustrate the layers used, their configurations, and the flow of data through the model.
Author action:
We added the architecture of the personalized CNN model on the manuscripts in section 5.2.
“Figure 4 Presents the architecture of the personalized CNN model.

Figure 4: the architecture of the personalized CNN model”
Comments 7 : Section 7.1, Experimental details, the author has not provided any information regarding the image type, disease type, modality type. It would be good to include few data statistics to know more on the input dataset used for the study.
Author response:
Thank you for your insightful feedback. We appreciate your attention to detail regarding the information provided in Section 7.1, Experimental details
We recognize the importance of including specific information about the image type, disease type, and modality type to provide a comprehensive understanding of the input dataset used for the study.
Author action:
In response to your comment, We added some details about the used datasets as follows:
Medical image datasets that include both images and textual descriptions, together with queries and ground truth. The majority of medical data sets currently available do not fulfill these criteria. Some sources lack assessment protocols, such as OHSUMED [51], while others focus on textual analysis and evaluation, like TREC. On the other hand, the ImageCLEFmed evaluation campaign offers specific medical picture collections for the purpose of assessing medical image retrieval. From 2011 onwards, the quantity and extent of the collections were comparable to those seen in real-world applications [ 59 ]. Due to copyright restrictions, the redistribution of the ImageCLEFmed collections to research groups is only allowed through a special agreement with the original copyright holders [52 ]. Therefore, we are restricted to conducting experiments using only the five collections for which we have obtained copyrights. The collections are shown in Table 1 and consist of two relatively small data sets: 74,902 and 77,495 images for the 2009 [53 ] and 2010 [54]
data sets, respectively. After the evolution of ImageCLEF, three additional data sets were added: 230,088 images for the 2011 [ 56 ] data set, and 306,539 images for both the 2012 [ 55] and 2013 [? ] data sets. Each image in these data sets is accompanied by a textual description. An image can have the text from its caption or a hyperlink to the HTML page that has the complete text of the article [ 58 ], together with the title of the article [53]. Furthermore, the queries were chosen from a collection of themes suggested by physicians and clinicians in order to precisely mimic the information requirements of a clinician involved in diagnostic tasks. The images in the 2009 and 2010 data sets were sourced from the RSNA journals Radiology and Radiographics [53 ] and consist of a portion of the Goldminer data set. Nevertheless, the photos in the data sets from 2011, 2012, and 2013 are derived from studies that were published in open-access journals and may be accessed through PubMed Central. The later data sets have a wider range of images, including charts, graphs, and other nonclinical images, resulting in more visual variety
Comments : Figure 4, the figure description looks incomplete and doesn’t provide enough information about the figure. It would be good to include figure description for better clarity.
Author response:
Thank you for highlighting the need for clarity. We have revised Figure 4 to improve clarity and added a detailed description. We tune the value of alpha on different datasets to evaluate its impact on the results using different metrics such as MAP, P@5, and P@10.
As explained in section 7.1, we have carried out our experiments using 4 datasets: ImageCLEF collections in Medical Retrieval Task for years 2009 , 2010, 2011 and 2012.
Author action:
We update the caption of Figure 4:
“Results according to α using 4 ImageCLEF datasets 2009, 2010, 2011 and 2012”
Comments : The results shown in the manuscript on the dataset are good, but it has not been validated on any clinical medical datasets. Also, it is unclear which modalities and image types are supported by this model.
Author response:
Thank you for your valuable feedback. We appreciate your suggestion to provide more details regarding the validation of our model on clinical medical datasets and the supported modalities and image types.
In response to your comments, we would like to clarify that our model has been validated using four different ImageCLEF datasets from the years 2009, 2010, 2011, and 2012. These datasets were created by experts in the medical domain and are widely recognized as benchmarks in the field of medical image retrieval. They are part of a global challenge that attracts participation from researchers working on similar problems, ensuring a high standard of quality and relevance.
Furthermore, we have added a detailed paragraph to our manuscript that describes the types of images and modalities included in these datasets. This information will help readers understand the scope and applicability of our model. Specifically, the datasets include a variety of medical imaging modalities, which cover a wide range of medical conditions and scenarios
Author action:
In response to your comment, as we explain in our answer to your comment number 7, we added some details about the used datasets as follows:
“Medical image datasets that include both images and textual descriptions, together with queries and ground truth……”
Comments: The metrics shown in the results section was accuracy, it would have been good if the authors would have used other metrics such as Sensitivity, specificity and F-1 score for evaluation.
Author response:
Thank you for your insightful feedback regarding the evaluation metrics used in our study.
We understand the importance of using a variety of metrics to provide a comprehensive evaluation of our model's performance. In the context of information retrieval, the primary focus is on the relevance of the retrieved items, especially within the top results. Therefore, precision at rank 5 (P@5), precision at rank 10 (P@10), and Mean Average Precision (MAP) are widely regarded as the most informative metrics for assessing performance in this field. These metrics directly measure the effectiveness of retrieving relevant images among the top results, which is crucial for practical applications.
We chose P@5, P@10, and MAP because they are well-suited for evaluating the relevance of retrieved images, particularly in medical image retrieval tasks where the goal is to find the most relevant images quickly. Also, We chose to use MAP (Mean Average Precision), P@5, and P@10 metrics because these are the standard metrics used in the ImageCLEF challenge, which allows for direct comparison between all participants.
We appreciate your suggestion and believe this addition will strengthen the robustness and applicability of our findings.
Thank you once again for your constructive feedback and valuable insights. We look forward to the opportunity to enhance our paper in line with your recommendations.
Best regards,

Round 2
Reviewer 1 Report
Comments and Suggestions for Authors
Most of comments have been considered by authors in the revised version. The proposed approach is described more clear than original submission in this version. So, the revised version is better than original submission in terms of paper organization and technical details.